# Evaluation of Mechanical Properties and Microscopic Structure of Coal Gangue after Aqueous Solution Treatment

**DOI:** 10.3390/ma12193207

**Published:** 2019-09-30

**Authors:** Yan Zhang, Xiaoyun Yang, Susan Tighe

**Affiliations:** 1College of Energy and Transportation Engineering, Inner Mongolia Agricultural University, Hohhot 010018, China; ycyangxiaoyun@outlook.com; 2Department of Civil & Environmental Engineering, University of Waterloo, Waterloo, N2L 3G1, Canada; sltighe@uwaterloo.ca; 3Yuncheng polytechnic college, Yuncheng 044000, China

**Keywords:** coal gangue, embankment filling, XRD, XRF, SEM, EDS

## Abstract

Coal gangue, a solid waste produced in coal production, had caused serious environmental pollution due to accumulation on dumps. Embankment filling can solve the problem while significantly consuming the amount of coal waste for mining. The main purpose of this study is to investigate the mechanical properties and microscopic structure of coal gangue when it is subjected to erosion from water environment with different acidity. Using immersion testing to evaluate its stability in different hydro-chemical environments. Mechanical property parameters of coal gangue treated by solutions were investigated. The action microstructure of coal gangue was revealed through a series of X-ray diffraction (XRD), X-ray fluorescence spectrometry (XRF), scanning electron microscopy (SEM), and energy dispersive spectrometry (EDS). The results show that acidic solution behaved better improvement effect on compressive modulus and fraction of coal gangue samples owing to the generation of quartz and the reduction of aluminum, dissolving of some substances, and transforming of small scattered angular grains through soaking treatment. Alkalinity treatment can be chosen to improve cohesion of coal gangue as a result of polymeric silicon aluminum salt, with high viscosity, was produced by chemical reaction during immersion. Therefore, aqueous solution treatment contributes to engineering properties and presents great potential in both supplement road building materials and recycling of coal gangue.

## 1. Introduction

The large-scale development of coal resources in China has produced a large amount of solid waste such as coal gangue in the process of coal production, which resulted in fearful detriment to coal mining and the surrounding areas [1,2,3]. It was reported that the amount of coal gangue accumulated in China has reached 4.5 billion tons, occupying more than 1.5 million ha of land, and causing serious environmental pollution [4,5]. Flammable gases from coal gangue spontaneous combustion cause different degrees of pollution to the environment such as atmosphere, water, soil, and so on [6,7,8]. In addition, if atmospheric precipitation seeps into the self-igniting coal gangue pile, the rapid expansion of heated air will lead to the risk of dump expansion or explosion [9]. The contradiction between the concern of resource depletion and environmental damage caused by fuel consumption is becoming more and more prominent [10,11]. Therefore, the harmless treatment and utilization of coal gangue has increasingly become the focus of attention of researchers at present. The application of soil and rock fillers is very expansive because of the cost and transportation. Coal gangue is mainly consisting of sandstone and limestone and is suitable as a filling material with high bearing capacity and simple operation process in the mining area [12]. The coal mining method of reclaimed land and embankment filling exemplifies the application and characteristics of green mining from resource recycling and meets the challenges of coal gangue dumps on the surface and the shortage of road construction materials. Various studies have been performed on the utilization of coal gangue in road construction and it is believed that coal gangue can be directly used for subgrade fillers [13,14]. However, due to complex geological and extreme climatic conditions, some coal gangues are subjected to long-term corrosion by water infiltration through the surface or groundwater. Coal gangues are susceptible to acidic or alkaline water owing to high organic matter [15,16,17,18,19]. Therefore, the interactions between water and coal gangue includes physical and chemical interactions, which have the potential to change the microstructure and mineral composition of coal gangue and reduce their mechanical properties. 

Most previous studies have focused on physical methods to improve the stability of coal gangue filling, such as controlling filling height, impact compaction, dynamic compaction, and material incorporation [20,21,22,23,24,25]; and chemical methods to increase the strength of the material by means of alkali-activated [26,27]. The mechanical properties of filling materials are one of the decisive factors in the mass of the filling quality [28,29] and acidity led to significant strength reduction in the backfilling [30]. After being filled, when encountering extreme water environmental conditions, the mechanical properties of coal gangue will alter dramatically, thus affecting the durability and the strength of the filling structure. Therefore, studies on the influence of water on the mechanical properties of coal gangue are significant for estimate the stability of coal gangue fillings. Acid, neutral and alkaline three solutions were used to soak coal gangue samples. The influence of acid and alkaline solution treatment on the properties of coal gangue subgrade filling was experimentally evaluated by XRD first, and then microstructure features were demonstrated by scanning electron microscope (SEM) and energy dispersive spectrometer (EDS) measurements. The main purpose of this research is aimed at using XRD, XRF, SEM, and EDS to analyze the strength change mechanism of coal gangue applied in subgrade filler after treatment through aqueous solutions. It is of great significant to increase bonging of coal gangue particles and ensure the embankment filling water stability. Simultaneously, to reduce the environmental pollution through consuming piled coal gangue on the surface.

## 2. Materials and Methods 

### 2.1. Sampling and Specimen Preparation

The coal gangue (CG) samples used in this study were collected from the Jinyang Coal Mine in Ordos, China (Figure 1). 

The coal gangue was first broken by a crusher, and then filtered by a classifying screen to obtain a group of samples with a particle size of 9.5–16 mm. The basic physical and mechanical parameters measured according to the test procedure [31] and ISO/DIS 20290-1 [32] as shown in Table 1. Based on the basic physical parameters of coal gangue, it is a kind of sandstone. The different acidic and alkaline static immersion solution were prepared with analytical purity, including CH_3_COOH, tap water, and NaOH. The initial pH formulated solutions are 4.5, 7.2, and 8.5, respectively. Solutions were added to the coal gangue at the solid–liquid ratio (mass ratio) of 1:1, and the immersion was performed for 15 days. Immersed coal gangue samples were selected for microstructure measurements, such as XRD, XRF, SEM, and EDS. 

### 2.2. Experiments Methods

The confined compressive tests (the tri-axial compressive tests under uniaxial strain condition) are commonly used to study the deformation properties of loose backfill materials [35,36,37]. Compressive modulus and shear strength indexes for every sample were measured through the compressive test and triaxial test in accordance with Chinese Standard JTG E40—2007 [38], ISO 22965-2:2007 and ISO 17892-8:2018 [39,40]. Triaxial testing carried by a Triaxial Apparatus (TSZ30—2.0, Nanjing T-Bota Scietech Instruments & Equipment Co., Ltd., Nanjing, China) with 0.5–1%/min shear strain rate. 

To characterize the chemical compositions and microstructures of samples, pH, XRD, XRF, SEM, and EDS tests were carried out. The pH value of samples was tested by a PHS-3C pH meter (Model PHS-3C, Shanghai INESA Scientific Instrument Co., Ltd., Shanghai, China). In order to characterize the phase composition and microstructure of the CG samples, 10 g of uniformly mixed was randomly selected for the following tests. XRD equipment used was an X-ray diffractometer (XRD-6000, Shimadzu Corporation, Kanagawa Japan), with Cu-Kα radiation (*a* = 0.154 nm) over a 2θ range from 10° to 40°. The generator voltage and electric current of 40 kV and 40 mA, continuous scanning speed of 2°/min, and sampling interval was 0.1°. The content of oxides and elemental components is measured by X-ray fluorescence spectrometer (Axios Pw4400, PANalytical B.V., Almelo, Netherlands), with rhodium target end window X-ray tube of 4 kW. Pre-spray platinum on the sample for SEM determination test. The microstructure observation was performed on samples using a scanning electron microscope (ZEISS^TM^ SUPRA 55, Carl Zeiss Co., Ltd., Jena, Germany) with an energy dispersive spectrometer which was operated at an acceleration voltage of 30kV. An energy dispersive spectrometer (X-Max80, Oxford Instruments Company, Oxford, England) with 80 mm^2^ detector crystal, 2000× amplification factor, and 129 eV resolution was selected to determine CG samples’ elements and chemical components.

## 3. Results and Discussions

### 3.1. Mechanical Properties

The basic mechanical properties of CG are very important to the design of embankments. Therefore, compressive modulus *E*, cohesion *c*, and fraction *φ*, were measured by compressive testing and unconsolidated undrained triaxial test under unsoaked and soaked conditions. In the experiments, a total of 24 samples were used and repeated three times per experiment, with the average value of the value of each mechanical parameter as the final value of each mechanical parameter (show in Table 2). 

Table 2 shows that the basic properties of the CG changed after the CG samples were treated by the acidic, neutral, and alkaline solution. To be specific, compressive modulus increased from 73.1 MPa under raw state to 158.3 MPa, 142.5 MPa, and 146.2 MPa—respectively, under acidity, neutral, and alkalinity conditions—increased by 116.6%, 94.9%, and 100%. In addition, their cohesion *c* dropped from 15.1 kPa in an untreated state to 8.0 kPa and 9.7 kPa, respectively, under immersed in acidity and neutral solutions, while increased to 71.3 kPa under soaked in alkaline solution. In terms of samples soaked in acidity, the friction angle φ rose 12.5%, while in neutral solutions, the internal friction angle *φ* almost no increase, nevertheless, dropped in alkaline solution. Consequently, the acidic solution contributes to compressive modulus and *φ*, while the alkaline solution is beneficial to increasing the compressive modulus and *c*. 

### 3.2. pH Values

The initial pH values of the solutions non-immersed and immersed coal gangue samples through 15 days are shown in Table 3.

The pH values of these solutions reached about 5.5 through soaking 15 days as shown in Table 3, conforming to requirements for water of concrete (higher than 4.0) specified by Chinese standard JGJ 63—2006 [41] and ISO 12439:2010 [42]. The sample soaked in acid solution decomposed and precipitated alkaline substances, which shows an increase in pH value. However, after soaking in neutral and alkaline solution, concentration of H^+^ ions decreased, resulting in a decrease in pH value.

### 3.3. Mineral Composition Analysis

The mineral composition of coal gangue affected the filling body’s strength. To identify the change in the mineral composition caused by the solutions with different acidities, an XRD diffractometer was used to measure the mineral diffraction map of the coal gangue. The mineral components of coal gangue before and after immersion are presented in Figure 2. 

According to XRD testing, the calculation results of substances in the coal gangue samples by Jade Software are shown in Table 4.

Comparative analysis with the standard powder diffraction data for various substances provided by the Powder Diffraction Federation International Data Center (JCPDS—ICDD), the hydro-chemical environment has significantly changed the mineral composition of the coal gangue (in Figure 2). The main characteristic peaks are located at 20°, 21°, 27°, 37°, and 39°. After soaking in acidic, neutral, and alkaline solutions, the diffraction peaks and peak shapes of CG samples changed slightly. The main mineral composition for every sample found by the k-value method of jade software are quartz (SiO_2_), kaolinite (Al_2_O_3_-2SiO_2_-2H_2_O), plagioclase (NaAlSi_3_O_8_-CaAl_2_Si_2_O_8_), illite, and mica. By comparing with the control standard sample-CG, raw-CG contain more quartz and less kaolinite. For raw-CG samples, the content of quartz is 45.4%. In terms of CG samples treated by acidic, neutral, and alkali solution, SiO_2_ of the sample increased by 11.7%, 17.6%, and 15.9%, respectively, mica, kaolinite, and plagioclase was all reduced, when compared with raw-CG. Some diffused peaks appear in the alkali-immersed sample at 2θ = 24°, 28°, and 33°. It could be an amorphous product silica-alumina gel according to the chemical composition of the sample [43]. 

In order to verify the analysis results of XRD, the XRF test method to detect the oxide composition for comparative analysis, the results of the analysis can be found in Table 5.

From Table 5, it can be seen that after immersion of acid–base solution, the content of SiO_2_ in the coal gangue has increased, while the relative reduction of oxides such as Al_2_O_3_, Fe_2_O_3_, CaO, etc., which proved the consistency of the results of the decomposition of mica, kaolinite, and plagioclase analyzed in XRD. 

### 3.4. Morphology Analysis

The morphology of CG samples with untreated and treated by various acidity solutions was investigated using XRF and SEM. Results are displayed in Table 6 and Figure 3.

As can be seen from Table 6, in raw-CG, pH = 4.5-CG, pH = 7.2-CG, and pH = 8.5-CG, Si/Al ratio are 1.36, 1.4, 1.2, and 1.39, respectively. The ratio of silicon aluminum has increased, and other soluble elements are relatively reduced, which is conducive to the stability of coal gangue.

It is shown in Figure 3, the surface of sample particles is rough and irregular. After acidic solution treatment (in Figure 3b), kaolinite and plagioclase dissociated into small unevenly distributed crystals with particle size from 20 µm to 40 µm and mainly small particles. The distribution of minerals in pH = 7.2-CG samples is relatively uneven, compared to that of pH = 4.5-CG samples, there is no significant dispersion between grains. Mineral particles in pH = 8.5-CG sample are evenly distributed, and among mineral particles were full of cotton-like amorphous gelatinous hydration substances, which produced during the soaking process so that surface morphology looks relatively dense which is consistent with the measurement result of *c*. 

### 3.5. Elements Analysis

The elemental composition in labelled regions (in Figure 3) were measured by EDS. The main elements content presented in Figure 4.

As shown in Figure 4, the peak strength of each element varies widely from sample to sample, and the content of each element are listed in Table 7.

According to Table 7, the proportion of C, Si, Al, O, in CG samples are higher than that of Fe, Ca, Mg, K, Na, and other elements. In raw-CG, pH = 4.5-CG, pH = 7.2-CG, and pH = 8.5-CG, Si/Al ratio is 1.6, 3.3, 2.2, and 3.5, respectively, proving that after the solution is soaked, the Al element is easier to dissolved in the solution, and the final generator is Si element as the main substance. The content of Si increased in the coal gangue samples and the other elements decreased after acid solution treatment. It could be concluded that a large amount of dispersed grains in SEM image are associated with an increasing in the Si content detected by EDS. The lower the content of alumina and decomposing substances in coal gangue, the higher its strength [13]. Therefore, the acid solution treatment is conducive to the formation of Si and reduction of Al, and after acid immersion, the large particulate matter in the coal zircon is dispersed into the small angular particles of each cluster, which increases the surface area of the particles, also enhancing the compressive modulus and *φ* of coal gangue. For alkali-soaked samples, EDS applied a certain depth to the gel material adhering to the coal gangue particles in Figure 3d in order to verify the diffuse peak material present in the XRD pattern and the gel material composition in the SEM. The surface sweep, combined with the elements in Table 7, was finally determined to be an aluminosilicate gel, which can adhere to the surface of coal gangue particles or fill inter-granular pores to obtain a more compact and stable structure, leading to a significant improvement of mechanical properties.

### 3.6. Effects of Solution Acidicity on the Mechnical Propeties of Coal Gangue

At the microscopic level, the changes caused by the interactions between the different acidity solutions and coal gangue result in changes in the mineral composition and mechanical properties of the samples. The reasons for these changes were determined from analyses based on the variety of the pH, ions dissolution from the coal gangue, and changes in the mineral composition of the samples.

After soaking in acetic acid solution, the CG sample contacts with acid solution to produce chemical reaction, which causes many metal cations to escape. Moreover, the original mineral chemical bonds are broken, resulting in the formation of new material SiO_2_ aggregates, forming uneven grains (Figure 3b). Because the acid substances in CG escaped from the CG samples after soaking in the water solution, which provided the acid water environment for the CG. In an acidic environment, kaolinite and plagioclase on the sample surface react with the H^+^ ions in the chemical solution were decomposed by acid, and the ion exchange of Al^3+^ and Ca^2+^ took place, leading to spaces between several fragments in the CG (Figure 3c). The reaction chemical Equations (1) and (2) are [44,45]
Al_2_O_3_·2SiO_2_·2H_2_O + 6H^+^ = 2Al^3+^ + 2SiO_2_ + 5H_2_O(1)
2Na_0.6_Ca_0.4_Al_1.4_Si_2.6_O_8_ + 1.4H_2_O + 2.8H^+^ = 1.4Al_2_Si_2_O_5_(OH)_4_ + 1.2Na^+^ + 0.8Ca^2+^ + 2.4SiO_2_(2)

As shown in Equation (1), the reaction of acid with kaolinite and plagioclase formed SiO_2_, Al^3+^, and Ca^2+^ exchanged by H^+^ ions. In XRD and EDS measurement results, it is consistent that the content of silica increases after treatment with acid solution. However, these of treatment with water and alkaline solution are different. That is, XRD shows an increase, but a decrease in EDS, which may be related to the difference points of EDS measurement. In addition, acidic solution is contributing to compressive modulus and *φ*. 

OH^−^ ions in alkali solution can break the Al–O–Al and Si–O–Si network on the surface of CG. [Si_4_]^4−^ and [AlO_4_]^5−^ formed, respectively, and then synthesized to a three-dimensional polymeric silica-alumina gel [46] according to Equations (3) and (4) [47].
n(SiO_5_·Al_2_O_2_) + 2nSiO_2_ + 4H_2_O→n(OH)_3_–Si–O–Al–O–Si–(OH)_3_(3)
n(OH)_3_–Si–O–Al–O–Si–(OH)_3_→(Na)-(–Si–O–Al–O–Si–) + 4H_2_O(4)

Thus, polymeric silicon aluminum salt gel can be formed in the gap of CG, so that other substances in CG are difficult to escape, so the viscosity of the sample increases (Figure 3d), and the density is improved. In immersion tests, Si^4+^ and Al^3+^ ion leaching depends on the surface reaction of OH^−^ aluminosilicate material, in other words, the OH^-^ ions in the alkali metal Na^+^ electrostatic interaction, electrostatic repulsion, and the Si^4+^ and Al^3+^ ion release. The surface of the coal gangue particle has more active Al_2_O_3_ and SiO_2_, which completely react with the solution OH^−^ ions. As a result, the surface of the active substance decreases as the reaction progresses further, and the reaction depth continues to increase [48]. 

According to Equations (3) and (4), silicon aluminum ions are connected to each other to form new materials to reduce the content of the determination, which is consistent with the results of XRD and EDS. Meanwhile, polymeric silicon aluminum salt gel, a very sticky substance started to bond coal gangue particles, generated by alkaline treatment. Therefore, resulted in the cohesion *c* of coal gangue samples treated with alkaline solution increased by 3.72 times.

## 4. Conclusions

Based on the above experimental results, the following conclusions can be obtained.
(1)From microstructure, the kaolinite and plagioclase in the coal-stone disintegrates into small particles with a particle size of 20–40 µm, after acidic solution treating, and the particles are relatively complete and partially separated in pH = 7.2-CG. However, relatively tight in surface of pH = 8.5-CG and the particles are filled with gels.(2)After solution treatment, the proportion of Si increases. The main producing substance treated with acid and water is SiO_2_, and that of silicate gels formed treated with alkaline solution.(3)The compressive moduli and *φ* of CG samples rose under the condition of treatment by acidity solution due to increasing of SiO_2_. In addition, the generator is multi-angled and larger than the surface area in pH = 4.5-CG contributed to *φ* of CG samples. Furthermore, c grew much when soaking in alkaline solution as a result of production of sticky substance polymeric silicon aluminum salt gel.

This research will provide an effective way to recycle coal gangue as a filler in embankment, which can consume a large amount of coal gangue to exhibit environmental benefits. Coal gangue, as a kind of hazardous material, should be treated by alkaline solution in order to enhance the adhesion and density between particles of coal gangue, and improve the resistance to external water environment. However, chemical analysis—such as heavy metal leaching test—requires further study to evaluate the environmental impact of coal gangue. 

## Figures and Tables

**Figure 1 materials-12-03207-f001:**
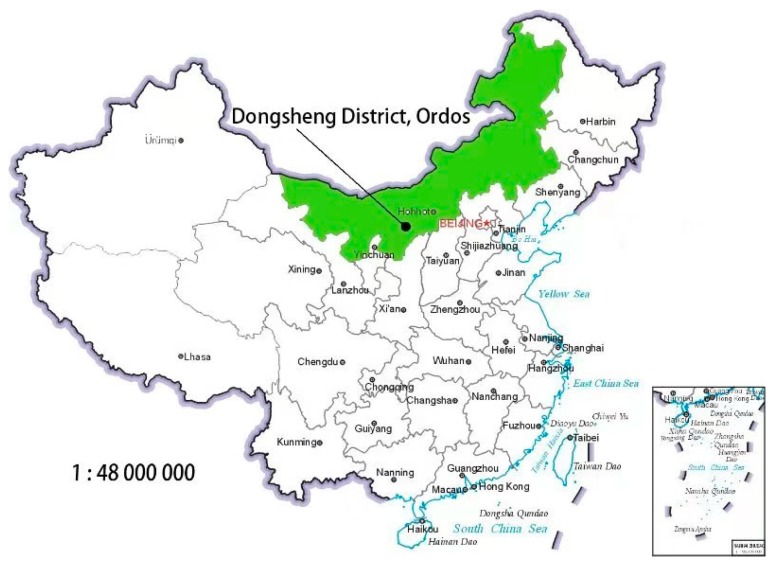
Location of CG samples in Ordos, China.

**Figure 2 materials-12-03207-f002:**
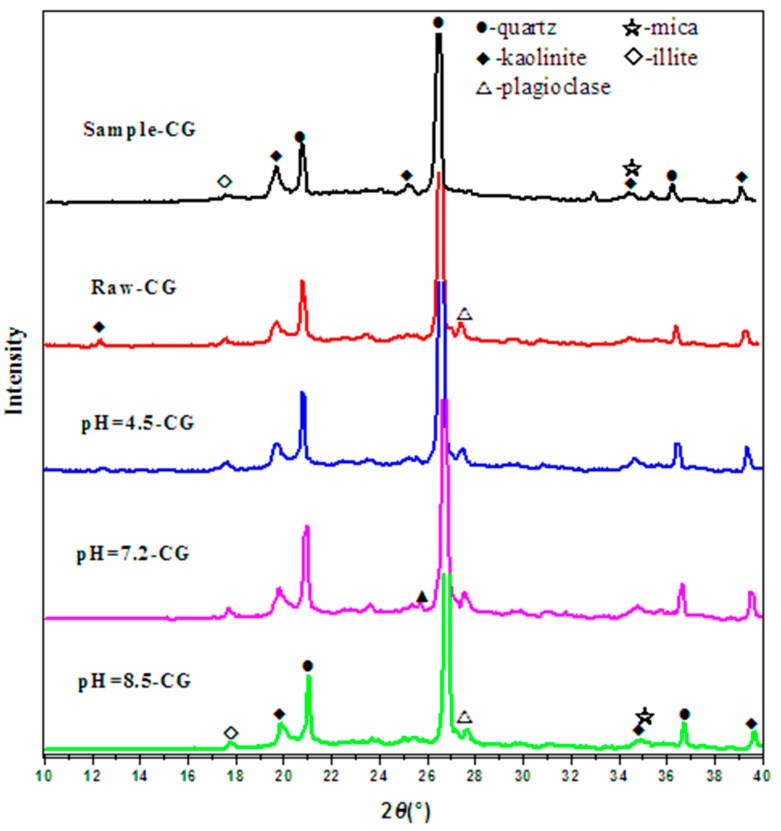
X-ray diffraction of CG samples. Note: Cu—Kα radiation wavelength radiation is 0.154 nm.

**Figure 3 materials-12-03207-f003:**
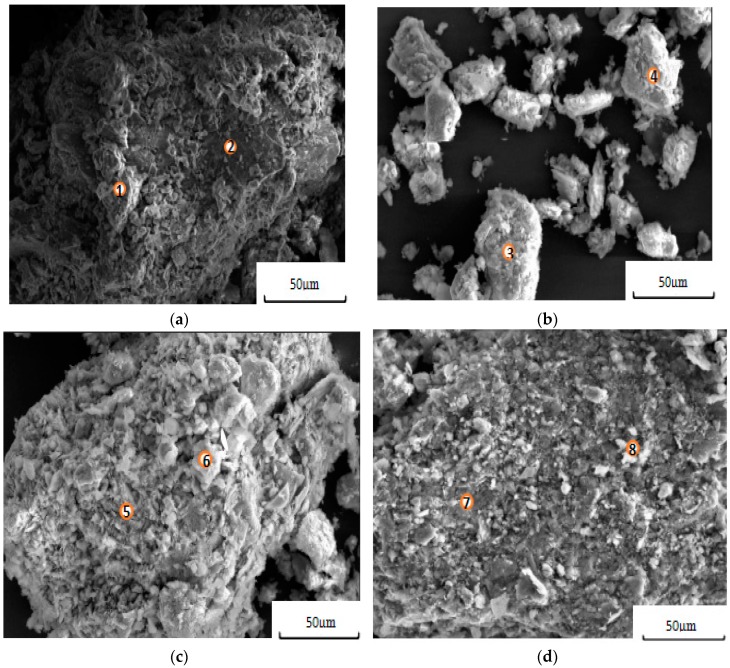
SEM images of coal gangues. (**a**) Raw-CG sample; (**b**) pH = 4.5-CG; (**c**) pH = 7.2-CG; (**d**) pH = 8.5-CG.

**Figure 4 materials-12-03207-f004:**
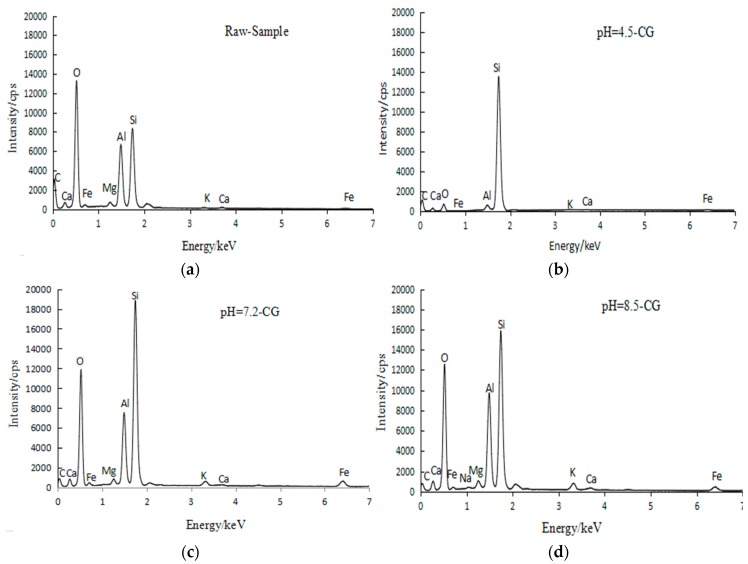
EDS spectrum in SEM images of coal gangue. (**a**) Raw-CG sample; (**b**) pH = 4.5-CG; (**c**) pH = 7.2-CG; (**d**) pH = 8.5-CG.

**Table 1 materials-12-03207-t001:** Basic physical parameters of coal gangue/%

Water Absorption	Water Solubility	Crushing Value	Consistence	Loss of Ignition [33]	Content of C [34]
3.7	4.7	24.2	31.9	18.6	3.6

**Table 2 materials-12-03207-t002:** Basic mechanical properties.

Sample Type	Compressive Modulus/MPa	Cohesion/kPa	Internal Friction Angle/°
Raw-CG	73.1	15.1	30.3
pH = 4.5-G	158.3	8.0	34.1
pH = 7.2-CG	142.5	9.7	30.4
pH = 8.5-CG	146.2	71.3	12.1

**Table 3 materials-12-03207-t003:** pH values of solutions.

Sample Type	Initial pH Value	After 15 Days Immersion
pH = 4.5-CG	4.5	5.4
pH = 7.2-CG	7.2	5.4
pH = 8.5-CG	8.5	5.6

**Table 4 materials-12-03207-t004:** Content of each substance in the coal gangue samples by XRD/%.

Sample Type	Quartz	Mica	Illite	Kaolinite	Plagioclase
Raw-CG	45.4	36.6	11.6	2	4.3
pH = 4.5-CG	50.7	31.4	13.2	1.8	3
pH = 7.2-CG	53.4	32.1	10.3	1.4	2.8
pH = 8.5-CG	52.6	31.6	11.4	1.5	2.8

**Table 5 materials-12-03207-t005:** Content of each oxide in the coal gangue samples by XRF/%.

Sample Type	SiO_2_	AL_2_O_3_	Fe_2_O_3_	CaO	MgO	SO_3_	K_2_O	P_2_O_5_	Na_2_O	TiO_2_
Raw-CG	45.8	32.8	4.5	1.1	7.7	0.6	4.1	0.2	1.2	2
pH = 4.5-CG	47.7	32	4.4	0.8	7.9	0.4	4.1	0.2	0.7	1.8
pH = 7.2-CG	45.6	32.9	4.1	1	8.7	0.5	4.1	0.3	1	1.8
pH = 8.5-CG	48.2	31.6	4.4	0.8	7.4	0.4	4.1	0.2	1.1	1.8

**Table 6 materials-12-03207-t006:** Content of each element by XRF/%.

Sample Type	Si	Al	O	Fe	Ca	S	Mg	K	Na	P	Ti
Raw-CG	26.2	19.2	37.3	3.9	1	0.3	5.1	4.3	1	0.1	1.6
pH = 4.5-CG	26.6	19.0	38.5	3.8	0.7	0.2	5.0	4.1	0.6	0.1	1.4
pH = 7.2-CG	20.8	17.1	48.1	2.8	0.7	0.2	5.1	3.3	0.7	0.1	1.1
pH = 8.5-CG	25.1	18.0	41.6	3.6	0.7	0.2	4.8	3.9	0.8	0.1	1.2

**Table 7 materials-12-03207-t007:** Average of the content of each element in the area measured/%.

Sample Type	C	Si	Al	O	Fe	Ca	Mg	K	Na	Others
Raw-CG	29.8	19.1	12.2	32.4	2.0	0.7	1.3	1.9	-	0.6
pH = 4.5-CG	37.8	22.5	6.8	31.6	0.6	0.1	0.3	0.2	-	0.1
pH = 7.2-CG	22.6	14.8	6.7	54.2	0.5	0.1	0.5	0.4	-	0.1
pH = 8.5-CG	21.2	15.6	4.1	57.5	0.4	0.1	0.4	0.4	0.3	-

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
