# Peer review of "Evaluation of Mechanical Properties and Microscopic Structure of Coal Gangue after Aqueous Solution Treatment"

_materials, 2019, doi:10.3390/ma12193207_

Round 1
Reviewer 1 Report
Review of the manuscript “Evaluation of mechanical properties and microscopic mechanism of coal gangue treated with different acidity aqueous solutions” submitted
to Materials by Yan Zhang, Xiaoyun Yang and Susan Tighe
Utilization of coal mining solid wastes is a problem of great concern. Studies of different properties of coal mining wastes are very important because of their variability and numerous potential applications.
In my opinion the submitted manuscript is not suitable for publication because of flaws in research planning, to general description of experiments and interpretation of results.
Detail comments:
Line 2-4. Title is not clear: „Evaluation of mechanical properties and microscopic mechanism of coal gangue treated with different acidity aqueous solutions”. What „microscopic mechanism” is considered?
Line 75-80: Description of preparation of solutions is unclear. Concentration of reagent is unknown. pH value of distilled water (7.2) seems to be unrealistic.
Line 130. Interpretation of decrease of pH values is unclear. pH value is related to concentration of H+ ions in solution!
Lines 132-152. Numerous comments: a. Results of XRD analyses are documented using diffractograms of very poor quality. b. For identification of minerals in coal solid wastes diffractograms should be recorded in broader range (e.g. from 5 to 65o2Q CuKa). It is almost impossible to identify mica minerals using narrow analytical range. c. What values are listed (e.g. d=1454)? d. Precise identification of anorthite is doubtful. I suggest to use „plagioclase”! e. Usage of Rietveld method is suggested for quantification of minerals content in samples. f. Quantification of content of Al2O3 used by Authors is unknown! What form of occurrence of Al2O3 is suggested? Crystalline or amorphous? How it is possible to determine content of amorphous Al2O3 is sample? g. Don’t use capital letters in mineral names.
Lines 154-166. Preparation of samples for SEM investigation is unknown. Probably small fragments were attached on holders and coated with carbon. Analysis of porosity is unrealistic under these conditions. Several methods of rocks porosity measurements are commonly applied. Description of figures (Figure 2) – „pores” instead of „Proes”
Lines 168-193. The chapter is based on unrealistic results. Single chemical analysis using EDS method in small spots gives information about chemical composition in this spot! Chemical composition of sample can be determined by averaging numerous spot analyses or, better by analysis of bigger areas.
What are evidences of the presence of Na2SiO3 in samples?
Lines 194-234. Considerations related to chemical transformation are doubtful and not supported by any citations.
English language in the manuscript is of poor quality.

Reviewer 2 Report
English should be reviewed-
Line 73: Include situation map
Lines 86, 128: indicate the equivalent international ISO standards.
Figure 1: Indicate the units in the ordinate axis.
Line 144: The chemical formula of Kaolinite is not the one indicated. In the mineralogical composition there are no amorphous materials?
Line 146: Chemical analysis indicated. How have they been measured?
Figure 2: Proes?
Line 194: acdicity ?, Gnague?
Line 218: The new materials, how have they been identified? Justify it.
Reference 26: does not adapt to the standards of the magazine.
Reviewer 3 Report
The topic of the paper is of interest regarding some environmental impacts that coal industry might produce and also the opportunities created by the wastes from this industry.
I think that the title should be rewritten.
The introductory section could benefit of more background regarding the utilisation of natural and waste-related fillers.
The expression "different acidic solutions" should be rephrased since the experiments used only one acidic solution, the other two were neutral and basic.
In order to use this type of waste in the construction industry, other tests should be addressed (heavy metal leaching; density; friction etc.), these other properties should be mentioned at least in conclusion as future work. Also, the conclusion should present other directions of the work and the limitations of the material towards using it in the construction industry.
Reviewer 4 Report
(1)In several points - for example in abstract, in paragraph 3.1- it is written internal fraction angle instead of friction, and the sentence has not meaning
(2) In Figure 2, it is written "proes" instead of pores
(3) In Table 2, the measurements standard deviation shall be added to check if the values are statistically significantly different.
(4) The conclusions shall be more analytical to understand better the results of the research
4.1 It is written: "This research ...provides an effective way to recycle coal gangue as a filler in embankment, exhibiting environmental benefits". The meaning of effective way is treatment with acidic or alkaline solutions? What of the two? What is the connection with the engineering practice? What are the environmental benefits?
4.2 It is written: "... should be treated by variable acidity solution ...". The authors propose acidic, neutral or alkaline solutions? Or combination? Or it depends on the application?
Round 2
Reviewer 1 Report
Comments to the revised version of the manuscript “Evaluation of mechanical properties and microscopic structure of coal gangue treated through aqueous solution” submitted to Materials
by Yan Zhang, Xiaoyun Yang and Susan Tighe
Authors have presented answers to all remarks included in the review.
Several corrections are insufficient. Chemical analysis based on EDS method can be presented as an example. In the first review it was suggested to determine chemical composition in numerous spots or areas and to calculate the average. It is important to remember that chemical composition of the rock can vary from one spot to another and single determination informs us about the chemical composition in this spot (additional remark: please include information about rock type in coal gangue, e.g. sandstone, mudstone). The best idea is to apply other method of chemical analysis (e.g. XRF) and to supplement results in determination of LOI (loss of ignition) value, content of C and S (e.g. using LECO). Interpretation of such results of chemical analysis will be less speculative.
Additional determination of the chemical composition of solutions after experiment will strongly support interpretation of chemical reactions during gangue-solution interaction.
Detail comments are included below:
Lines 2-4: I suggest modification of the title, e.g. „Evaluation of mechanical properties and microscopic structure of coal gangue after aqueous solution treatment”
Line 14: “open accumulation”? May be “accumulation on dumps”?
Line 36: “environment” instead of “natural environment”
Line 44: “low transportation cost” – cost of transportation is related to the distance.
Line 52: “acid-alkaline”? at the same time? Is it possible that high organic matter content is responsible both for formation of alkaline and acid solutions?
Line 65: “….acid and alkaline solution…. ” instead of “acid-alkali”
Line 84: How it is possible to obtain pure water of pH 7.2? Usually the reaction with atmospheric CO2 lowers pH value. What mean “pure water”? Distilled water? Tap water?
Line 104: What means: “Pre-pray platinum on the sample for SEM determination”?
Line 119-120: How standard deviation was calculated? Standard deviation calculated for all results for all samples (raw-CG, pH=4.5-CG, pH=7.2-CG, pH=8.5-CG)? What means this parameter?
Line 143-144: “X-ray diffractometer” instead of “X-ray spectrometer”.
Line 145-146: Interpretation of X-ray diffractions patterns: a. method of distinction between illite and mica? b. I suggest to remove values of intensity from the Figure, c. Legend to horizontal axis – please add information about radiation wavelength, e.g. “o2Q CuKa”, d. How to identify mica-type minerals (mica and illite) without first order reflections (ca 10Å).
Line 160: “the content of quartz is 45.4%, which gave the sample a rather high hardness”. Hardness of the sedimentary rock is not directly related to the quartz content. Type of cement is usually more important.
Line 163: Total content of minerals listed in table 4 for sample Raw-CG is 99.9%. How to understand statement: “In addition, the content of Al2O3 in the 163 Raw-CG is 10.4%”. Total content is over 110%? Why Al2O3 is not marked in diffractograms?
Line 166: How it is possible to reduce content of Al2O3 by 11.7, 17.6 and 15.9%? Content in starting material is 10.4%! It should be explained in more clear way.
Line 167: “broad peaks”, “diffused peaks” instead of “diffusion peaks”
Line 178: Photos are not the illustration of the porosity. Fig. 3b presents several fragments of the rock and spaced between them are not pore space in the rock.
Line 187: Chemical composition was “analysed”, “measured” but not “confirmed”!
Line 197: Contents of chemical elements correspond to analytical spots not to samples! To measure chemical composition of the sample using EDS method we need to analyse chemical composition of several areas and next to calculate an average value. Values presented in Table 5 cannot be used in interpretation.
Line 198: Table 4 or Table 5?
Line 200: What is the source of data related to silica content? “… content of silica in Raw-CG, pH=4.5-CG, pH=7.2-CG and pH=8.5-CG is 44.5%, 66.7%, 29.3% and 22.9%....”
Line 219: “chemical bonds are broken, decomposed and dispersed”. What is decomposition and dispersion of chemical bonds? May be “broken” is enough?
Line 222-223: “In an acidic environment, kaolinite and plagioclase on the sample surface react with the H+ ions in the chemical solution were decomposed by acid, and the ion exchange of Al3+ and Ca2+ took place, resulting in some voids in the CG (Figure 3(c)).” The sentence is unclear! Voids in Fig. 3c represent spaces between several fragments.
Lines 228-250: Discussion is poorly supported by results presented in the manuscript.

Reviewer 2 Report
Is OK
Author Response
No need to modify.